# Cullin-RING Ubiquitin Ligases in Neurodevelopment and Neurodevelopmental Disorders

**DOI:** 10.3390/biomedicines13040810

**Published:** 2025-03-28

**Authors:** Honoka Ashitomi, Tadashi Nakagawa, Makiko Nakagawa, Toru Hosoi

**Affiliations:** 1Department of Clinical Pharmacology, Faculty of Pharmaceutical Sciences, Sanyo-Onoda City University, Sanyo-Onoda 756-0084, Japan; p120003@ed.socu.ac.jp (H.A.);; 2Division of Cell Proliferation, United Centers for Advanced Research and Translational Medicine, Graduate School of Medicine, Tohoku University, Sendai 980-8575, Japan; 3Institute of Gene Research, Yamaguchi University Science Research Center, Ube 755-8505, Japan; 4Advanced Technology Institute, Life Science Division, Yamaguchi University, Ube 755-8611, Japan

**Keywords:** Cullin-RING ubiquitin ligase, neurodevelopment, neurodevelopmental disorders

## Abstract

Ubiquitination is a dynamic and tightly regulated post-translational modification essential for modulating protein stability, trafficking, and function to preserve cellular homeostasis. This process is orchestrated through a hierarchical enzymatic cascade involving three key enzymes: the E1 ubiquitin-activating enzyme, the E2 ubiquitin-conjugating enzyme, and the E3 ubiquitin ligase. The final step of ubiquitination is catalyzed by the E3 ubiquitin ligase, which facilitates the transfer of ubiquitin from the E2 enzyme to the substrate, thereby dictating which proteins undergo ubiquitination. Emerging evidence underscores the critical roles of ubiquitin ligases in neurodevelopment, regulating fundamental processes such as neuronal polarization, axonal outgrowth, synaptogenesis, and synaptic function. Mutations in genes encoding ubiquitin ligases and the consequent dysregulation of these pathways have been increasingly implicated in a spectrum of neurodevelopmental disorders, including autism spectrum disorder, intellectual disability, and attention-deficit/hyperactivity disorder. This review synthesizes current knowledge on the molecular mechanisms underlying neurodevelopment regulated by Cullin-RING ubiquitin ligases—the largest subclass of ubiquitin ligases—and their involvement in the pathophysiology of neurodevelopmental disorders. A deeper understanding of these mechanisms holds significant promise for informing novel therapeutic strategies, ultimately advancing clinical outcomes for individuals affected by neurodevelopmental disorders.

## 1. Introduction

Ubiquitination is a dynamic post-translational modification that plays critical roles in maintaining cellular function and homeostasis by modulating protein stability, activity, and localization [1,2]. Ubiquitination refers to the covalent attachment of ubiquitin molecules to a substrate protein. This process is carried out through a sequential enzymatic cascade involving three key enzymes. First, the E1 ubiquitin-activating enzyme activates ubiquitin in an ATP-dependent manner, forming a high-energy thioester bond. Next, the activated ubiquitin is transferred to the E2 ubiquitin-conjugating enzyme. Finally, the E3 ubiquitin ligase facilitates the transfer of ubiquitin from the E2 enzyme to the target protein, determining the substrate specificity [3]. The transfer of ubiquitin to the substrate is orchestrated by either a RING (really interesting new gene) finger domain, a HECT (homologous to the E6-AP carboxyl terminus) domain, or an RBR (RING-between-RING) domain, leading to the classification of ubiquitin ligases into RING-type, HECT-type or RBR-type based on the domain they possess [4,5]. Ubiquitin ligases function as either single-protein entities which contain both E2- and substrate-binding domains, or multi-subunit complexes which harbor a protein responsible for binding E2, and a substrate receptor protein.

The human and rodent genomes encode more than 600 ubiquitin ligases, which play pivotal roles in various cellular processes, including proliferation, differentiation, and apoptosis [6]. These ligases are implicated in numerous pathophysiological conditions, such as cancer, immune dysfunctions, aging, and developmental disorders [7]. The largest subclass of ubiquitin ligases is the Cullin-RING ubiquitin ligase (CRL), in which Cullin protein serves as a scaffold to bind both the RING domain protein ROC1/2 (also called RBX1/2), and the substrate receptor protein directly or indirectly through an adaptor protein [8,9]. Mammalian cells contain eight (CUL1, 2, 3, 4A, 4B, 5, 7, 9) Cullin proteins, each of which builds their specific ubiquitin ligase complexes [8,9]. For example, CUL1-based ubiquitin ligase uses F-box proteins as a substrate receptor which is linked to CUL1 via a SKP1 adaptor protein [10], while CUL4A or CUL4B employs DDB1 adaptor protein which recruits DCAF substrate receptor proteins [11,12] (Figure 1). In contrast to CUL1, CUL2, CUL3, CUL4A, CUL4B, and CUL5, recent structural investigations have demonstrated that CUL7 and CUL9 directly interact with their substrates, with the ubiquitylation module being supplied by CUL1 in the case of CUL7 [13]. CUL9, on the other hand, employs its intrinsic RBR domain to help catalyze ubiquitylation [14]. All Cullin proteins are thought to use ROC1 as a RING domain protein, except for CUL5 which uses ROC2 [15] (Figure 1). In addition to these Cullin-based complexes, APC/C (anaphase promoting complex/cyclosome) ubiquitin ligase constitutes a similar structure in which APC2 serves as a main scaffold, APC11 as a RING domain protein, and CDC20 or CDH1 as a substrate receptor [16] (Figure 1). Due to this structural similarity and its significant functions in neurodevelopment [17], we include and discuss APC/C ubiquitin ligase in this review.

Ubiquitination has been demonstrated to play a significant role in neurodevelopmental processes such as neural stem cell proliferation/differentiation, axonal/dendritic growth, migration, synaptogenesis, and synaptic function (Figure 2). Thus, dysregulation of these processes has been linked to several neurodevelopmental disorders (NDDs), including autism spectrum disorder (ASD), intellectual disability (ID), and attention-deficit/hyperactivity disorder (ADHD). Genomic studies have identified both CRLs and other E3 ligases, including HECT-type E6-AP (also known as UBE3A), SMURF1, and RBR-type ARIH2, as being linked to NDDs [18,19,20,21,22], underscoring the need for comparative analysis and potential interaction between CRLs and other ubiquitin ligases in neurodevelopment. This review aims to explore the role of CRLs-catalyzed ubiquitination in neurodevelopment and its implications for NDDs, highlighting their potential as therapeutic targets to bridge the gap between fundamental research and clinical applications. Readers are referred to excellent reviews on ubiquitin ligases and neurodevelopment for a more comprehensive understanding of ubiquitination in neurodevelopment, focusing on areas such as immune dysregulation [23], axon development [24]. and spatiotemporal regulation of HECT-type ubiquitin ligases [25].

## 2. Human Genetics of CRL in NDD Patients

Exome sequencing and chromosomal microarray testing of NDD patients identified de novo heterozygous loss-of-function mutations in *CUL3* on chromosome 2 [26,27], homozygous or compound heterozygous loss-of-function mutations in *CUL7* on chromosome 6, and hemizygous loss-of-function mutations in *CUL4B* on X chromosome [28,29] in NDD patients. Furthermore, variants of genes encoding CRL1 substrate receptors have been implicated in NDD patients. These include FBXO10 [30], FBXO11 [31,32], FBXO28 [32,33], FBXO31 [34], FBXO47 [35], FBXL3 [36], FBXL4 [37,38], FBXL10 (also known as KDM2B) [39], β-TrCP1 (also known as FBXW1) [40,41,42], β-TrCP2 (also known as FBXW11) [41], and FBXW7 [43,44,45]. Additionally, variants in CRL3 substrate receptors, such as KLHL15 [46], KLHL17 (also known as actinfilin) [21], KLHL20 [47], KCTD7 [48], and KCTD13 [49] are also relevant. Likewise, variants in CRL4 substrate receptors, including DCAF1 (also known as VprBP) [50], DCAF14 (also known as PHIP, BRWD2 and RepID) [51,52], COP1 [53], and CRBN [54,55] have been implicated in these patients. Lastly, mutations in gene encoding APC/C substrate receptor CDH1 [56] have also been associated with NDD. These findings suggest the involvement of these genes in NDDs, thereby prompting further research to elucidate how these mutations contribute to the pathogenesis of NDDs.

## 3. Behavioral Phenotypes of Mice with CRL Mutations

Mouse models have been developed to investigate the mechanisms underlying the pathogenesis of NDDs [57,58,59]. To assess the suitability of these models for NDD research, various behavioral evaluations have been designed and conducted [60,61]. For instance, social interaction deficits are quantified through open field and three-chamber social interaction tests, while impairments in learning and memory are assessed using a range of maze and contextual fear conditioning paradigms [60,61]. Additionally, increased distance traveled serves as an indicator of hyperactivity, and anxiety is primarily evaluated through open field and elevated plus-maze tests [60,61]. Several CRL mutant mouse models have been shown to exhibit these NDD-associated behaviors, as summarized in Table 1. These findings suggest that mutations in these genes are most likely causal, rather than merely correlative, in the etiology of NDDs.

## 4. CRLs in Neural Stem Cell Proliferation and Differentiation

In the following sections, we will provide an overview of the current understanding of the molecular mechanisms underlying neurodevelopment and NDDs, as derived from studies on CRLs.

Neural stem cells (NSCs) are multipotent progenitor cells that give rise to neurons, astrocytes, and oligodendrocytes during brain development [72,73]. NSCs first give rise to radial glial cells (RGCs), which serve dual roles as neural progenitors and as scaffolds that guide migrating neurons. RGCs divide asymmetrically, producing either neurons directly or intermediate progenitors (IPs) as an intermediary step in neurogenesis. IPs undergo a limited number of divisions, amplifying the production of neurons to meet developmental demands. Mature neurons are generated from the differentiation of IPs or directly from RGCs in a highly regulated process. This sequential progression from NSCs to RGCs, then to IPs, and finally to mature neurons ensures the proper formation and organization of the nervous system, which is essential for the precise development of neural structures and circuits [74,75,76]. The tightly regulated processes of NSC proliferation and differentiation rely heavily on the dynamic control of protein expression and degradation. CRLs play a pivotal role in these processes by regulating key signaling pathways and transcription factors (Figure 3).

NSC proliferation is governed by the precise regulation of cell cycle progression, which is controlled by cyclins, cyclin-dependent kinases (CDKs), and their inhibitors. Ubiquitination ensures the timely degradation of these cell cycle regulators, thus maintaining proper cell cycle transitions. For example, APC/C ubiquitin ligase recognizes mitotic regulators such as cyclin B through the CDH1 substrate receptor (APC/C^CDH1^), and targets them for ubiquitination, promoting the progression of mitosis [77,78]. In *Cdh1*-deficient neural stem cells, aberrantly elevated cyclin B-associated CDK activity induces DNA damage-mediated apoptotic cell death [78], potentially by disrupting the precise regulation of S phase timing, thereby eliciting DNA replication stress [77]. Another key regulator is the CUL1-based ubiquitin ligase complex (CRL1, also known as SKP1-CUL1-F-box SCF ubiquitin ligase), which ubiquitinates cyclin E for degradation with the use of the FBXW7 substrate receptor (CRL1^FBXW7^). ASPM suppresses this SCF^FBXW7^-mediated ubiquitination, thereby stabilizing cyclin E. This stabilization facilitates the shortening of the G1 phase in the cell cycle, promoting NSC proliferation [79].

NSC differentiation requires the fine-tuned suppression of self-renewal programs and the activation of lineage-specific transcriptional networks. Ubiquitination modulates this process by targeting signaling pathway components and transcription factors that drive differentiation. For instance, the Notch signaling pathway, which maintains NSC and RGC self-renewal, is also regulated by the ubiquitin ligase CRL1^FBXW7^, which ubiquitinates and targets Notch intracellular domain (NICD) for proteasomal degradation [80]. This degradation facilitates the transition from NSC maintenance to astrocytic differentiation [80]. APC/C also plays a critical role in NSC differentiation. APC/C^CDH1^ ubiquitinates and facilitates the degradation of the CDK activator CDC25A and SKP2, a substrate receptor of SCF ubiquitin ligase that targets CDK inhibitors for degradation [78]. This process prevents cell cycle re-entry, thereby promoting neuronal differentiation [78]. Similarly, the Wnt/β-catenin signaling pathway, a critical regulator of NSC maintenance, is modulated by ubiquitination. The E3 ligase CRL1^β-TrCP^ ubiquitinates β-catenin, leading to its degradation and attenuation of Wnt signaling [81,82].

In addition to cell cycle and signaling pathways, ubiquitination directly regulates transcription factors critical for NSC identity and differentiation. SOX2, a core transcription factor that maintains NSC pluripotency, is subject to ubiquitination. In human pluripotent stem cell-derived NSC, CUL4A-based Cullin-RING ubiquitin ligase with DET1-COP1 heterodimer as a substrate receptor (CRL4A^DET1-COP1^) ubiquitinates SOX2 for degradation, leading to neural differentiation [83].

The dysregulation of ubiquitination in NSC proliferation and differentiation has profound implications for NDDs. Mutations in genes encoding CRL-related proteins (please refer to Section 2) and FBXW7 regulator ASPM [84,85] are identified in NDD patients and these could be caused by aberrant NSC proliferation of differentiation.

## 5. CRLs in Neuronal Polarization

Immature neurons extend processes known as neurites, which subsequently differentiate into axon, the intracellular signal-transmitting structure, or dendrites, the primary signal-receiving components of neurons. The developmental process that defines this distinction is known as neuronal polarization, signifying the establishment of distinct subcellular compartments to enable precise signal transmission and reception [86]. Neuronal polarization is initiated by extracellular symmetry-breaking cues, followed by cytoskeletal rearrangements [87], with actin filaments (F-actin) guiding axon/dendrite extension and microtubules playing an essential role in axon stabilization [88,89]. By modulating protein functions, CRLs ensure the spatial and temporal coordination of neuronal polarization (Figure 4).

The protein kinase AKT facilitates axon formation by phosphorylating GSK-3β. This phosphorylation negates the GSK-3β’s function, leading to the activation of microtubule-binding proteins and subsequent stabilization of microtubules [90]. PTEN is a negative regulator of AKT activity [91], necessitating PTEN inactivation within the axon. Notably, the HECT-type ubiquitin ligase NEDD4 is reported to facilitate PTEN degradation through ubiquitination, thereby promoting neurite outgrowth [92] and axon branching [93]. However, the decrease in ubiquitination and accumulation of PTEN were not detected in NEDD4-deficient neurons [94], indicating that the E3 ubiquitin ligase for PTEN in developing axon awaits to be revealed. Although not in neuronal cells, CRL4B^DCAF13^ [95,96] and SCF^FBXO22^ [97] are reported to ubiquitinate PTEN for degradation, providing the possibility that Cullin-RING ubiquitin ligases are involved in axon formation through PTEN regulation.

The activity of a small G protein RhoA is critical for dendrite formation through promoting actin arc formation, which prevents microtubule protrusion and axon growth [98]. RhoA is stabilized in dendrites by the ubiquitin ligase APC/C^CDH1^ that ubiquitinates the RhoA destabilizing HECT-type ubiquitin ligase SMURF1 for degradation [99]. In the axon, the activity of RhoA is downregulated by a mechanism in which RhoA activator PDZ-RhoGEF is targeted for degradation by the ubiquitin ligase CUL3 with KLHL20 serving as a substrate receptor (CRL3^KLHL20^) [100]. A symmetry-breaking signaling by BDNF induces the phosphorylation of PDZ-RhoGEF, potentiating its ubiquitination and subsequent degradation [100], facilitating neuronal polarization.

The interplay between F-actin and microtubules, mediated by coupling proteins, is also implicated in axon and dendrite formation [101]. One such coupling protein, DCX, is subject to negative regulation by CRL3^KLHL15^ [102], CRL4A^CRBN^ [103], and CRL4B^CRBN^ [103]. The ubiquitination and subsequent proteasomal degradation of DCX by these CRLs have been shown to attenuate axonal and dendritic complexity and length [102,103].

An additional regulatory layer in the process of neuronal polarization is provided by transcriptional mechanisms, whose activities are also modulated by ubiquitination. For instance, APC/C^CDH1^ facilitates the ubiquitination of SnoN [104], a transcription factor that promotes the expression of positive regulators of axon growth, and ID2 [105], an inhibitor of axon growth repressors, thereby suppressing axon extension. We have also demonstrated that the ubiquitin ligase CUL4B regulates NGF-induced neurite extension via transcriptional modulation [106,107]. CUL4B mediates the ubiquitination and subsequent degradation of WDR5 [106], a key component of the histone H3 lysine 4 methyltransferase transcriptional complex [108], in rat neuroblastoma PC12 cells, downregulating the neuronal gene expression [109]. These findings underscore the critical role of transcriptional regulation in neuronal process formation.

Dysregulation of ubiquitination in neuronal polarization also has profound implications for NDDs. Mutations in genes encoding CRL-related proteins (please refer to Section 2), as well as DCX [110,111], SMURF1 [21,22], and PTEN [112,113] are identified in NDD patients and these could be caused by aberrant neuronal polarization.

## 6. CRLs in Neuronal Migration

NSCs are generated, and remain, adjacent to the brain ventricle, while differentiating neurons migrate towards the cortical surface, with later-differentiated neurons passing by earlier-generated neurons during corticogenesis [76,114]. This inside-out layering of the cortex is guided by various mechanisms, some of which are regulated by CRLs (Figure 5).

The most well-characterized regulator of neuronal migration is Reelin/DAB1 signaling [115]. Reelin binds to its receptors and activates tyrosine kinases, leading to the phosphorylation of DAB1 [116]. Phosphorylated DAB1 then interacts with several proteins, including LIS1, a regulator of microtubule-based dynein-dynactin motor proteins, which facilitates cytoskeletal reorganization essential for migration [117]. CUL5 with SOCS proteins as substrate receptors (CRL5^SOCS^) targets phosphorylated DAB1 for ubiquitination and degradation, thus preventing DAB1 hyperactivation [118]. CUL5 knockdown-induced accumulation of phosphorylated DAB1 results in excessive migration and abnormal superficial positioning [118]. Furthermore, CUL4B was shown to bind to LIS1 [119], suggesting that CRL4B may also play a role in regulating neuronal migration through Reelin signaling.

In contrast, the heterozygous deletion of *Cul3*, a condition that mirrors NDD in humans, leads to impaired migration of embryonic neurons and cortical lamination abnormalities in mice [63]. Proteomic analysis has identified PLS3, an actin-bundling protein, as a critical substrate of CUL3, and the accumulation of PLS3 caused by *Cul3* haploinsufficiency disrupts actin filament organization and results in abnormal adhesion with impaired migration [63]. The substrate receptor responsible for PLS3 recognition remains unidentified.

The generation of the cerebellum and hippocampus also depends on proper neuronal migration [120,121]. FBXO41, a neuron-specific substrate receptor of CRL1 ubiquitin ligase, has been shown to promote neuronal migration in the cerebellum (from the external to the internal granule layer) [122], and hippocampus (from the hilus to the granule cell layer) [123]. It localizes to the centrosome and disassembles primary cilia, an antenna-like structure that receives several signaling ligands [124]. However, the involvement of primary cilia and the substrates of SCF^FBXO41^ in neuronal migration remain unclear.

In addition to CUL3 and CUL4B, mutations in genes encoding Reelin [125,126], DAB1 [127], and LIS1 [128] have been identified in patients with NDDs, indicating that defects in neuronal migration may, at least in part, contribute to the pathogenesis of these disorders.

## 7. CRLs in Synaptogenesis and Synaptic Function

Elongating axons extend towards the dendrites of target neurons, where synapses are established [129,130]. At the presynaptic terminal, neurons secrete neurotransmitters such as glutamate, dopamine, acetylcholine (ACh), and gamma-aminobutyric acid (GABA), which bind to their corresponding receptors in the specialized postsynaptic density (PSD) region of the postsynaptic neurons [131], thereby enabling communication and the transmission of information. CLRs play a significant role in synaptogenesis and synaptic function (Figure 6).

Presynaptic development is transcriptionally regulated by APC/C^CDC20^. In this process, the ubiquitination and degradation of the transcription factor NeuroD2 lead to the downregulation of Complexin II expression. Complexin II acts as a negative regulator of presynaptic differentiation, and its downregulation promotes the formation of presynaptic structures [132]. Both the presynaptic active zones and PSDs are supported by scaffolding proteins, the abundance of which is frequently regulated by ubiquitin ligases [133]. For example, FBXL20 (also known as SCRAPPER) localizes at the presynaptic terminal, where it targets RIM1, a Ca^2+^-sensing synaptic vesicle regulator, for ubiquitination and degradation, ensuring the proper release of glutamate [134].

Glutamate stimulates postsynaptic neurons by activating four types of ion channels—AMPA, kainate, NMDA, and GluD receptors [135]—and the G-protein-coupled metabotropic receptor (mGluR) [136]. Chronic synaptic activity triggers the degradation of the AMPA receptor, a mechanism critical for preventing the toxic hyperactivation of postsynaptic neurons, known as excitotoxicity. APC/C^CDH1^ plays a pivotal role in the downregulation of the AMPA receptor by targeting the GluA1 (also known as GluR1) subunit for ubiquitination and degradation in response to synaptic stimuli [137]. Periodic synaptic activation induces long-term potentiation (LTP) or long-term depression (LTD) mediated through glutamate receptors, which are the foundations of synaptic plasticity essential for learning, memory, and other processes frequently dysregulated in NDD patients [138]. APC/C^CDH1^ is involved in mGluR-mediated LTD by synaptic stimulus-dependent ubiquitination of FMRP1, a negative regulator of LTD, leading to its degradation [139]. Accordingly, the loss of CDH1 in excitatory neurons of the mouse brain impairs LTD, resulting in sustained cell surface expression of the AMPA receptor and defective degradation of FMRP1 [139]. The levels of the NMDA receptor are also regulated by CRL3^KCTD13^ which targets GluN1 for ubiquitination and degradation [140]. In patients with epileptic seizures, the expression of KCTD13 is diminished, likely leading to the heightened activation of excitatory synaptic transmission and an increased susceptibility to epilepsy [140].

Dysfunction of the dopamine system is implicated in the phenotypes associated with NDDs [141,142]. Consistently, mice with a heterozygous *Cul3* deletion in DA neurons exhibit hyperactivity of DA neurons accompanied by the behavioral abnormalities such as increased locomotion, impaired working memory, and deficits in sensorimotor gating (please refer to Table 1), all of which were reversed by the forced inactivation of the DA neuron [65]. The hyperexcitability of DA neurons was shown to be caused by the accumulation of HCN2 channels, a substrate of CUL3 [65]. These findings confirm the critical role of elevated DA activity in NDDs.

Potassium channels are essential for maintaining neuronal activity by regulating membrane potential [143]. The potassium channel Kv10.1 (also known as EAG1 or KCNH1), whose gene is mutated in NDDs [144,145], undergoes ubiquitination-mediated degradation, a process driven by CRL7^FBXW8^ [146]. Consequently, the overexpression of CUL7 diminishes potassium currents, whereas reduced CUL7 expression leads to enhanced potassium currents in non-neuronal cells that exogenously express Kv10.1 [146]. The extent to which these findings can be extrapolated to neuronal function requires further investigation.

The postsynaptic sites in dendrites often form specialized protruded structures known as spines [147]. KLHL17 has been shown to enlarge spines and facilitate synaptic activity. Since this function requires the BTB domain, which is essential for association with CUL3, CRL3^KLHL17^ likely contributes to this process, although the specific substrate remains to be identified [67]. Dendrites are dynamic structures that must be maintained to ensure the proper functioning of the brain. One destabilizing factor of dendrites is ROCK2, whose levels are regulated by APC/C^CDH1^ [71]. Therefore, the loss of CDH1 leads to the accumulation of ROCK2, resulting in dendritic spine disruption and learning deficits, which can be rescued by a ROCK inhibitor [71].

## 8. Therapeutic Implications and Future Directions

The critical roles of Cullin-RING ubiquitin ligases (CRLs) in neurodevelopment and their involvement in NDDs, as summarized in Table 2, highlight the potential for therapeutic targeting. Given that mutations in genes encoding CRL-related proteins have been identified in NDD patients (as discussed in Section 2), modulating ubiquitination pathways represents a promising approach to mitigate disease phenotypes. One potential therapeutic strategy involves the use of small-molecule inhibitors to regulate CRL activity. For instance, MLN4924 (also known as pevonedistat) is a NEDD8-activating enzyme (NAE) inhibitor and disrupts all CRL functions, leading to the stabilization of CRL substrates [148]. MLN4924 has been demonstrated to be brain-permeable [149], and to alleviate ischemic brain injury [150]. Consistently, MLN4924 can prevent neuronal cell death induced by oxidative stress in hippocampal neurons and SH-SY5Y neural cells [151]. Furthermore, MLN4924 enhances the proliferative capacity and inhibits differentiation in corneal stem cells, thereby accelerating corneal epithelial wound healing [152]. These findings suggest the potential of MLN4924 as a therapeutic agent for NDDs associated with impaired neural stem cell proliferation or excessive cell death. MLN4924 has already entered clinical trials for cancer therapy [153], and the results from these trials will provide valuable insights into its adverse effects and clinical tolerability. In addition to a global CRL inhibitor, the targeted inhibition of specific CRLs may offer enhanced therapeutic potential for NDDs, necessitating the development of highly selective inhibitors aimed at specific CRL complexes or substrate receptors.

Not all of the CRL-related proteins discussed in this review have been identified as mutated, but the increasing sample sizes may reveal mutations in these genes. In relation, patient-specific genetic and molecular profiling can guide therapeutic interventions for NDDs. Whole-genome, transcriptome, and proteomic analyses could aid in identifying individual CRL-related mutations, enabling the selection of personalized therapeutic strategies. In addition, patient-derived induced pluripotent stem cells (iPSCs) offer an invaluable platform for modeling disease mechanisms and testing novel therapeutics in a personalized context [154].

Despite significant progress in understanding the role of CRLs in neurodevelopment, several key questions remain unanswered. Future research should focus on: (1) identifying substrate specificity; many CRL substrate receptors remain uncharacterized. Understanding their target specificity is essential for developing selective therapeutic interventions: (2) developing selective CRL modulators; current pharmacological tools lack specificity for individual CRL complexes, though CUL-level inhibitors are emerging, as exemplified by DI-1548 and DI-1859 for CUL3 [155], and 33-11 and KH-4-43 for CUL4 [156]. Advances in structure-based drug design may enable the development of highly selective CRL inhibitors or activators [157]: (3) reconciling the conflicting findings; for instance, as shown in Table 1, mice with a whole-body heterozygous knockout of the *Cul3* gene do not exhibit an anxiety-like phenotype [62,63], while those with a heterozygous knockout of *Cul3* in neural progenitor cells do exhibit an anxiety-like phenotype [64]. Furthermore, the reduction in Cul4b shortens neurite extension in PC12 cells [106], whereas *Cul4b* knockout does not affect dendritic length in hippocampal neurons [66]. Interestingly, the knockdown of Cul4b in cultured cerebral neurons increases the length of total neurites, axons, and dendrites [103]. Clarifying the factors contributing to these inconsistencies could shed light on the default functions of CRLs and the underlying causes of phenotypic heterogeneity often observed in NDD patients [158,159]. (4) Exploring non-degradative ubiquitination; while ubiquitination is often associated with protein degradation, non-degradative ubiquitin modifications also play critical roles in signaling pathways [160]. Investigating these roles could reveal novel therapeutic targets. (5) Investigating the role of CRLs in human tissues; a major limitation of current neurodevelopmental models is their reliance on rodent models. Human and rodent brains exhibit notable differences, such as fewer outer RGCs [161], and a reduced variety of interneuron cell types [162] in rodents. Human neurons differentiated from patient-derived iPSCs are being utilized to study neuronal functions [163], but these neurons lack the context of the brain’s native environment. Although studying the human brain raises ethical concerns, the development of human brain organoid techniques provides researchers with access to human brain tissue in vitro [164]. Pioneering studies are already revealing key functions of NDD-associated genes in differentiating human neural cells using brain organoids [165,166,167]. The functions of CRLs can similarly be investigated in the context of human brain development, which is crucial for obtaining more disease-relevant insights into CRL functions.

## 9. Concluding Remarks

The ubiquitination pathway, particularly through the action of CRLs, plays a fundamental role in neurodevelopment and the pathophysiology of NDDs. As our understanding of ubiquitin-mediated neurodevelopmental regulation continues to expand, new opportunities for treating NDDs are likely to emerge, bringing us closer to precision medicine-based interventions for these complex disorders.

## Figures and Tables

**Figure 1 biomedicines-13-00810-f001:**
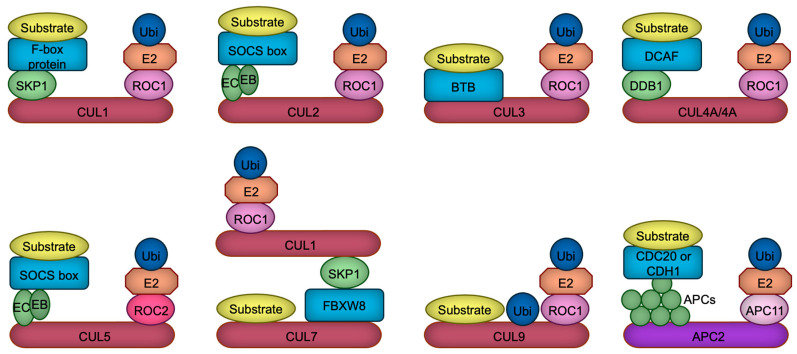
The composition of CRL ubiquitin ligases. Cullin proteins (red and purple) serve to link the substrate binding module with the ubiquitylation module. CUL1, CUL2, CUL4A, CUL4B, CUL5, and APC2 interact with substrate receptors (blue) through adaptor proteins (green), whereas CUL3 directly associates with substrate receptors. In contrast, CUL7 and CUL9 appear to bypass substrate receptors, binding directly to their substrates. CUL7 recruits CRL1 as a ubiquitylation module, whereas CUL9 uniquely harbors an RBR domain, which facilitates the catalysis of ubiquitylation through a ROC1-based ubiquitylation module. Ubiquitin, the E2 enzyme, the linker connecting the E2 enzyme to Cullin, and the substrate are represented in dark blue, orange, light pink, and yellow, respectively. EB, Elongin B; EC, Elongin C; APCs, APC proteins.

**Figure 2 biomedicines-13-00810-f002:**
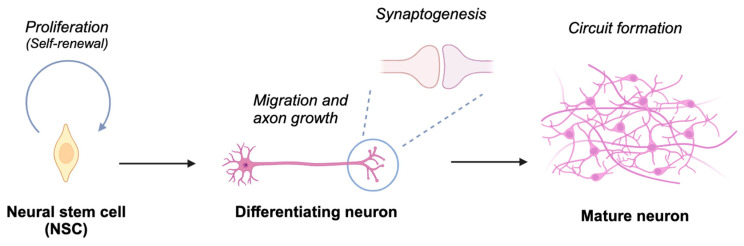
Schematic depiction of neuronal differentiation processes. Neural stem cells undergo proliferation and self-renewal. Upon initiating differentiation, these cells develop axons and establish synaptic connections with other neurons, culminating in the assembly of neural circuits essential for processing information in response to both environmental and endogenous stimuli. Created in BioRender. Nakagawa, T. (2025) https://BioRender.com/t69e513 (accessed on 5 March 2025).

**Figure 3 biomedicines-13-00810-f003:**
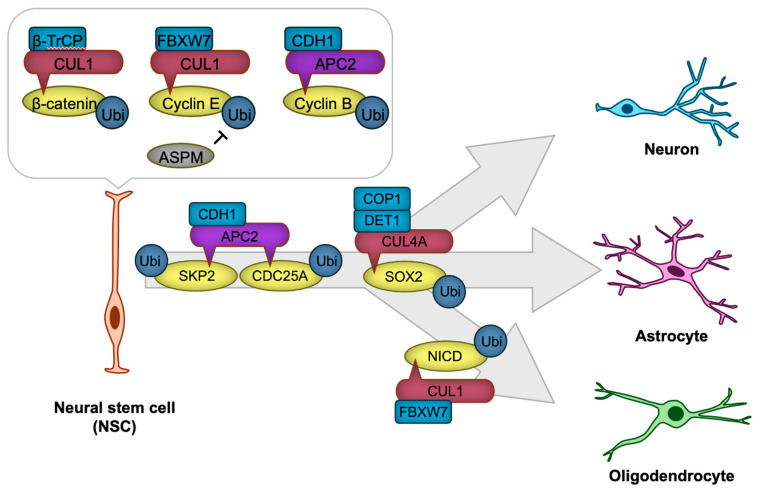
CRLs in neural stem cell (NSC) proliferation and differentiation. The transcription factor β-catenin and cyclins are targeted for degradation by CRL1 or APC/C to regulate NSC proliferation. NSC differentiation is precisely regulated through CRL1, CRL4A, or APC/C-mediated degradation of key cell cycle regulators, including SKP2 and CDC25A, as well as transcription factors such as SOX2 and NICD, thereby ensuring the timely generation of neurons or glial cells. The ‘-|’ symbol represents an inhibitory interaction.

**Figure 4 biomedicines-13-00810-f004:**
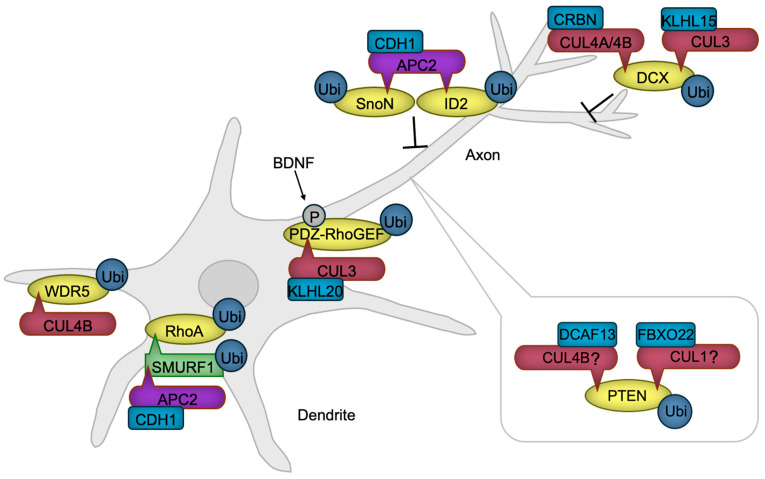
CRLs in neuronal polarization. APC/C promotes dendrite formation by targeting SMURF1, a negative regulator of RhoA, along with transcription factors SnoN and ID2, positive regulators of axon formation, for ubiquitylation and subsequent degradation. RhoA, essential for dendrite formation, is also regulated through the degradation of its activator, PDZ-RhoGEF, by CRL3. BDNF enhances this degradation, thereby accelerating axon formation. CUL4B-mediated degradation of WDR5 further promotes neurite growth. PTEN, a negative regulator of axonogenesis, is maintained at low levels in axons through ubiquitylation-mediated degradation. While the specific ubiquitin ligases remain unidentified, CRL4B and CRL1 are known to ubiquitylate PTEN in non-neuronal cancer cells, suggesting that these ligases may also regulate PTEN in developing neurons. DCX, which facilitates axonal and dendritic complexity, is negatively regulated by CRL3 and CRL4A/B. BDNF, Brain-derived neurotrophic factor. The ‘-|’ symbol represents an inhibitory interaction.

**Figure 5 biomedicines-13-00810-f005:**
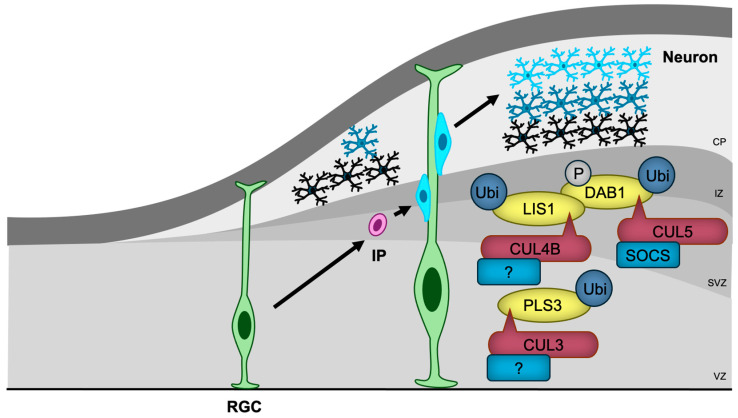
CRLs in neuronal migration. RGCs, the first differentiated cells derived from neural stem cells, generate IPs that migrate towards the cortical surface, with later-differentiated neurons passing by earlier-generated neurons. CRL3, CRL4B, and CRL5 regulate neuronal migration by ubiquitylating key migration regulatory proteins, including PLS3, LIS1, and DAB1, respectively. The substrate receptors of CUL3 and CUL4B for PLS3 and LIS1, respectively, have yet to be identified. RGC, radial glial cell; IP, intermediate progenitor; VZ, ventricular zone; SVZ, subventricular zone; IZ, intermediate zone; CP, cortical plate.

**Figure 6 biomedicines-13-00810-f006:**
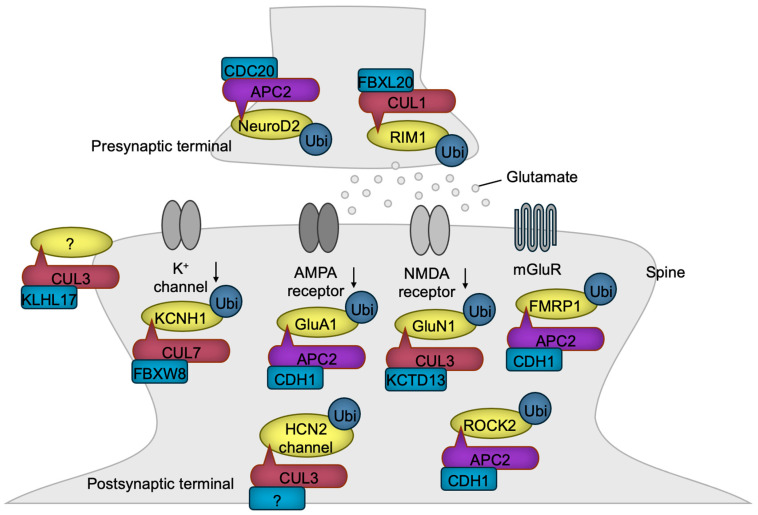
CRLs in synaptogenesis and synaptic function. CRL1 and APC/C regulate presynaptic terminal functions, while CRL3, CRL7, and APC/C control postsynaptic functions through the ubiquitylation of the indicated substrates. Notably, KLHL17-loaded CRL3 has been shown to promote spine enlargement and enhance synaptic activity, although the specific substrate responsible for this function remains unidentified. The substrate receptor of CUL3 for HCN2 channel has yet to be identified. The black downward arrows denote inactivation.

**Table 1 biomedicines-13-00810-t001:** NDD-related behavioral abnormalities exhibited by CRL-related gene mutations in mice. A check mark denotes the presence, while a minus sign in parentheses indicates the absence of the behavioral abnormalities. *, these mice exhibit hypoactivity. N.T., not tested; DA, dopaminergic; Glu, glutamatergic.

Target Disrupted		Behavioral Abnormality				Reference
Gene	Cell Type	Social Interaction	Learning and Memory	Hyperactivity	Anxiety	
*Cul3*	Whole body	✔	✔	✔	(−)	[62,63]
*Cul3*	Neurons and astrocytes	✔	(−)	(−)	✔	[64]
*Cul3*	DA neurons	N.T.	✔	✔	(−)	[65]
*Cul4b*	Whole body	(−)	✔	(−)	(−)	[66]
*Klhl17*	Whole body	✔	N.T.	✔	(−)	[67]
*Kctd13*	Whole body	(−)	✔	(−)	(−)	[68]
*Crbn*	Forebrain Glu neurons	N.T.	✔	(−)	(−)	[69]
*Cdh1*	Whole body	N.T.	✔	N.T.	N.T.	[70]
*Cdh1*	Forebrain Glu neurons	N.T.	✔	(−) *	✔	[71]

**Table 2 biomedicines-13-00810-t002:** CRL scaffolds, substrate receptors, their substrates, and the functions of ubiquitination discussed in this article. *, CUL4B does not require substrate receptor to bind to WDR5. NSC, neural stem cell; LTD, long-term depression.

CRL Scaffold	Substrate Receptor	Substrate	Function	Reference
APC/C	CDH1	Cyclin B	Promotion of NSC proliferation	[78]
CUL1	FBXW7	Cyclin E	Inhibition of NSC proliferation	[79]
CUL1	FBXW7	NICD	Promotion of NSC differentiation	[80]
APC/C	CDH1	CDC25A	Promotion of NSC differentiation	[78]
APC/C	CDH1	SKP2	Promotion of NSC differentiation	[78]
CUL1	β-TrCP	β-Catenin	Promotion of NSC differentiation	[82]
CUL4A	DET1-COP1	SOX2	Promotion of NSC differentiation	[83]
APC/C	CDH1	SMURF1	Promotion of dendritogenesis	[99]
CUL3	KLHL20	PDZ-RhoGEF	Inhibition of axon formation	[100]
CUL3	KLHL15	DCX	Inhibition of axon/dendrite complexity	[102]
CUL4A	CRBN	DCX	Inhibition of axon/dendrite complexity	[103]
CUL4B	CRBN	DCX	Inhibition of axon/dendrite complexity	[103]
APC/C	CDH1	SnoN	Inhibition of axon formation	[104]
APC/C	CDH1	ID2	Inhibition of axon formation	[105]
CUL4B	(−) *	WDR5	Promotion of neurite extension	[106]
CUL5	SOCS	DAB1	Inhibition of neuronal migration	[118]
CUL3	Unknown	PLS3	Promotion of neuronal migration	[63]
APC/C	CDC20	NeuroD2	Promotion of presynapse formation	[132]
CUL1	FBXL20	RIM1	Inhibition of postsynaptic function	[134]
APC/C	CDH1	GluA1	Inhibition of excess synaptic activity	[137]
APC/C	CDH1	FMRP1	Promotion of LTD	[139]
CUL3	KCTD13	GluN1	Inhibition of excess synaptic activity	[140]
CUL3	Unknown	HCN2	Inhibition of excess synaptic activity	[65]
CUL7	FBXW8	Kv10.1	Inhibition of potassium current	[146]
CUL3	KLHL17	Unknown	Promotion of synaptic activity	[67]
APC/C	CDH1	ROCK2	Destabilization of dendritic spines	[71]

## Data Availability

No new data were created or analyzed in this study.

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
