# Peer review of "Cullin-RING Ubiquitin Ligases in Neurodevelopment and Neurodevelopmental Disorders"

_biomedicines, 2025, doi:10.3390/biomedicines13040810_

Round 1

Reviewer 1 Report

Comments and Suggestions for Authors

The manuscript discusses the role of Cullin-RING ubiquitin ligases (CRLs) in neurodevelopment and neurodevelopmental disorders (NDDs). This is an important topic given the increasing evidence linking ubiquitin ligases to neurodevelopmental processes and disorders. The review compiles a comprehensive body of literature, synthesizing knowledge across multiple molecular pathways, genetic findings, and potential therapeutic implications. There is a focus on CRLs' involvement in processes such as neural stem cell proliferation, neuronal polarization, migration, and synaptic function - key aspects of neurodevelopment.

Strengths:

The review is well-structured, progressing logically from background information on ubiquitination to specific roles of CRLS in different aspects of neurodevelopment. The figures, including schematics of CRL composition and functions in neural development are complementary to the text and enhance comprehension.

The explanations of CRLs' roles in different neurodevelopmental processes are supported by experimental evidence.

Weaknesses:

Some sections, such as those discussing therapeutic implications, could benefit from a more in-depth critical analysis rather than a listing of potential strategies. Moreover, the claims made about therapeutic potential, particularly CRISPR-based interventions, are speculative. Any speculation should be reduced unless it can be supported by the literature. An area that could be strengthened is a discussion on conflicting findings or limitations of current models. It may be beneficial to include more discussion on how CRL dysfunction compares to other ubiquitin ligases implicated in neurodevelopment.

Minor comments:

  • using the term "communication" to describe behavioral results from open field and three chamber paradigms is misleading. Also, open field tests are usually interpreted as a measure of anxiety-like behaviors (as stated). It is recommended to change this descriptor to something more relevant, such as "social interaction".
  • some sentences contain long, complex sentences that could be simplified for better readability.
  • there are minor grammatical errors and awkward phasings (e.g. "Ubiquitination is dynamic post-translational modifications..." should be "Ubiquitination is a dynamic post-translational modification...")
  • While detailed, some descriptions are overly technical and could be simplified for clarity. 
Comments on the Quality of English Language

Overall, the quality of the language is good. However, addressing minor language issues and adding a more critical discussion of therapeutic challenges would improve the impact of this review.

Author Response

We sincerely appreciate your thorough review of our manuscript. Your comments have been invaluable in improving the quality of our work. Below, we provide detailed responses and outline the corresponding revisions, which are highlighted in the resubmitted files.

Comments and Suggestions for Authors

Comment:The manuscript discusses the role of Cullin-RING ubiquitin ligases (CRLs) in neurodevelopment and neurodevelopmental disorders (NDDs). This is an important topic given the increasing evidence linking ubiquitin ligases to neurodevelopmental processes and disorders. The review compiles a comprehensive body of literature, synthesizing knowledge across multiple molecular pathways, genetic findings, and potential therapeutic implications. There is a focus on CRLs' involvement in processes such as neural stem cell proliferation, neuronal polarization, migration, and synaptic function - key aspects of neurodevelopment.

Strengths: The review is well-structured, progressing logically from background information on ubiquitination to specific roles of CRLS in different aspects of neurodevelopment. The figures, including schematics of CRL composition and functions in neural development are complementary to the text and enhance comprehension.

The explanations of CRLs' roles in different neurodevelopmental processes are supported by experimental evidence.

Response: Thank you for the positive comment.

Comment: ”Weaknesses: Some sections, such as those discussing therapeutic implications, could benefit from a more in-depth critical analysis rather than a listing of potential strategies. Moreover, the claims made about therapeutic potential, particularly CRISPR-based interventions, are speculative. Any speculation should be reduced unless it can be supported by the literature. An area that could be strengthened is a discussion on conflicting findings or limitations of current models. It may be beneficial to include more discussion on how CRL dysfunction compares to other ubiquitin ligases implicated in neurodevelopment.

Response: We have revised the discussion on therapeutic implications to include a more in-depth critical analysis, as shown below, and removed the discussion on CRISPR-based strategies. In addition, we have added discussions on conflicting findings and the limitations of current models. Since we could not find literature comparing the dysfunction of CRLs and other ubiquitin ligases in neurodevelopment, we mentioned this gap in the introduction.

Added sentences related to in-depth therapeutic implication (line 404-415):

“MLN4924 has been demonstrated to be brain-permeable [149], and to alleviate ischemic brain injury [150]. Consistently, MLN4924 can prevent neuronal cell death induced by oxidative stress in hippocampal neurons and SH-SY5Y neural cells [151]. Furthermore, MLN4924 enhances the proliferative capacity and inhibits differentiation in corneal stem cells, thereby accelerating corneal epithelial wound healing [152]. These findings suggest the potential of MLN4924 as a therapeutic agent for NDDs associated with impaired neural stem cell proliferation or excessive cell death. MLN4924 has already entered clinical trials for cancer therapy [153], and the results from these trials will provide valuable insights into its adverse effects and clinical tolerability. In addition to a global CRL inhibitor, targeted inhibition of specific CRLs may offer enhanced therapeutic potential for NDDs, necessitating the development of highly selective inhibitors aimed at specific CRL complexes or substrate receptors.”

Added sentences related to conflicting findings (line 438-447):

“(3) reconciling the conflicting findings: For instance, as shown in Table 1, mice with a whole-body heterozygous knockout of the Cul3 gene do not exhibit an anxiety-like phenotype [62, 63], while those with a heterozygous knockout of Cul3 in neural progenitor cells do exhibit an anxiety-like phenotype [64]. Furthermore, the reduction of Cul4b shortens neurite extension in PC12 cells [106], whereas Cul4b knockout does not affect dendritic length in hippocampal neurons [66]. Interestingly, the knockdown of Cul4b in cultured cerebral neurons increases the length of total neurites, axons, and dendrites [103]. Clarifying the factors contributing to these inconsistencies could shed light on the default functions of CRLs and the underlying causes of phenotypic heterogeneity often observed in NDD patients [158, 159].”

Added sentences related to limitations of current models (line 450-461):

“(5) investigating the role of CRLs in human tissues: A major limitation of current neuro-developmental models is their reliance on rodent models. Human and rodent brains exhibit notable differences, such as fewer outer RGCs [161], and a reduced variety of inter-neuron cell types [162] in rodents. Human neurons differentiated from patient-derived iPSCs are being utilized to study neuronal functions [163], but these neurons lack the context of the brain's native environment. Although studying the human brain raises ethical concerns, the development of human brain organoid techniques provides researchers with access to human brain tissue in vitro [164]. Pioneering studies are already revealing key functions of NDD-associated genes in differentiating human neural cells using brain or-ganoids [165-167]. The functions of CRLs can similarly be investigated in the context of human brain development, which is crucial for obtaining more disease-relevant insights into CRL functions.”

Added sentences related to the comparison between CRLs and other ubiquitin ligases in neurodevelopment (line 87-90):

“Genomic studies have identified both CRLs and other E3 ligases, including HECT-type E6-AP (also known as UBE3A), SMURF1, and RBR-type ARIH2, as being linked to NDDs [18-22], underscoring the need for comparative analysis and potential interaction between CRLs and other ubiquitin ligases in neurodevelopment.”

Comment: “Minor comments:

  • using the term "communication" to describe behavioral results from open field and three chamber paradigms is misleading. Also, open field tests are usually interpreted as a measure of anxiety-like behaviors (as stated). It is recommended to change this descriptor to something more relevant, such as "social interaction".
  • some sentences contain long, complex sentences that could be simplified for better readability.

there are minor grammatical errors and awkward phasings (e.g. "Ubiquitination is dynamic post-translational modifications..." should be "Ubiquitination is a dynamic post-translational modification...")

  • While detailed, some descriptions are overly technical and could be simplified for clarity. 

Response: We thoroughly checked the manuscript and revised the indicated parts accordingly.

Comments on the Quality of English Language

Comment: “Overall, the quality of the language is good. However, addressing minor language issues and adding a more critical discussion of therapeutic challenges would improve the impact of this review.

Response: We hope that the revised version satisfactorily addresses the comments.

Reviewer 2 Report

Comments and Suggestions for Authors

Known in the field based on previous literatures:

  1. Protein ubiquitination is an important posttranslational regulation mechanism that mediates various pathways and function depending upon type and site of ubiquitination.
  2. Among the components of ubiquitin pathway, E3 ubiquitin-protein ligase is the most variable and divergent within eukaryotes and targeting specific E3 ligases would be a promising approach for molecular target for drug discovery.

In this review authors reported following findings:

I have gone through the review titled ‘Cullin-RING Ubiquitin Ligases in Neurodevelopment and Neurodevelopmental Disorders’. Authors mentioned and discussed about current knowledge on the molecular mechanisms of Cullin-RING ubiquitin ligases, the largest subclass of ubiquitin ligases, in the neurodevelopment and pathophysiology of neurodevelopmental disorders.

Authors described various CRL and their role in neuronal differentiation, polarization, behavioral abnormalities, and other neurodevelopmental processes.

The review article presented are interesting and generally supportive of the conclusions drawn. There are, however, several issues which need authors attention. The following suggestions if incorporated could help in the better understanding of the significance of the work and implications.

Minor/Major Concerns:

  1. Line 46, you have written BRB type. What is BRB type? It should be RBR.
  2. Line 78-80, in this sentence, authors can remove- In neurodevelopment, and rephrased it.
  3. Authors should elaborate introduction part bit more and they should describe about current number of E3 ligases, their role and involvement in different pathophysiological condition.
  4. Several review articles on E3 ligases are already available. Explain, how your review article is different from rest and how does it address a specific gap in the field?

Author Response

We sincerely thank you for your constructive comments, which have greatly contributed to the refinement of our manuscript. Detailed responses to your suggestions are provided below, with corresponding revisions highlighted in the resubmitted files.

Comments and Suggestions for Authors

Comments: “Known in the field based on previous literatures:

  1. Protein ubiquitination is an important posttranslational regulation mechanism that mediates various pathways and function depending upon type and site of ubiquitination.
  2. Among the components of ubiquitin pathway, E3 ubiquitin-protein ligase is the most variable and divergent within eukaryotes and targeting specific E3 ligases would be a promising approach for molecular target for drug discovery.

In this review authors reported following findings:

I have gone through the review titled ‘Cullin-RING Ubiquitin Ligases in Neurodevelopment and Neurodevelopmental Disorders’. Authors mentioned and discussed about current knowledge on the molecular mechanisms of Cullin-RING ubiquitin ligases, the largest subclass of ubiquitin ligases, in the neurodevelopment and pathophysiology of neurodevelopmental disorders.

Authors described various CRL and their role in neuronal differentiation, polarization, behavioral abnormalities, and other neurodevelopmental processes.

The review article presented are interesting and generally supportive of the conclusions drawn. There are, however, several issues which need authors attention. The following suggestions if incorporated could help in the better understanding of the significance of the work and implications.”

Response: Thank you for the positive comment.

Comments: “Minor/Major Concerns:

  1. Line 46, you have written BRB type. What is BRB type? It should be RBR.
  2. Line 78-80, in this sentence, authors can remove- In neurodevelopment, and rephrased it.
  3. Authors should elaborate introduction part bit more and they should describe about current number of E3 ligases, their role and involvement in different pathophysiological condition.
  4. Several review articles on E3 ligases are already available. Explain, how your review article is different from rest and how does it address a specific gap in the field?”

Response: We thoroughly checked the manuscript and revised the indicated parts accordingly. For comments 3 and 4, we have added the following sentences.

Added sentences related to the current number of E3 ligases, their role, and their involvement in pathophysiological condition. (line 50-53):

“The human and rodent genomes encode more than 600 ubiquitin ligases, which play pivotal roles in various cellular processes, including proliferation, differentiation, and apoptosis [6]. These ligases are implicated in numerous pathophysiological conditions, such as cancer, immune dysfunctions, aging, and developmental disorders [7].”

Added sentences related to the difference of this manuscript from others and how it addresses a specific gap in the field. (line 90-97):

“This review aims to explore the role of CRLs-catalyzed ubiquitination in neurodevelopment and its implications for NDDs, highlighting their potential as therapeutic targets to bridge the gap between fundamental research and clinical applications. Readers are referred to excellent reviews on ubiquitin ligases and neurodevelopment for a more comprehensive understanding of ubiquitination in neurodevelopment, focusing on areas such as immune dysregulation [23], axon development [24]. and spatiotemporal regulation of HECT-type ubiquitin ligases [25].”.